# INSYDE-content: a synthetic, multi-variable flood damage model for household contents

Pradeep Acharya<sup>1,2</sup>, Mario Di Bacco<sup>3</sup>, Daniela Molinari<sup>2</sup>, Anna Rita Scorzini<sup>1</sup>

- <sup>1</sup> Department of Civil, Environmental and Architectural Engineering, University of L'Aquila, 67100 L'Aquila, Italy
- <sup>2</sup> Department of Civil and Environmental Engineering, Politecnico di Milano, 20133 Milano, Italy
- <sup>3</sup> Department of Civil and Environmental Engineering, University of Florence, 50139 Firenze, Italy

Correspondence to: Anna Rita Scorzini (annarita.scorzini@univaq.it)

Abstract. This paper introduces INSYDE-content, a novel, probabilistic, multi-variable synthetic model designed to estimate flood damage to household contents on a component-by-component basis. The model addresses a critical gap in current modeling tools, which often overlook the significance of household contents in overall damage assessments. Developed through an expert-based approach and grounded in the scientific and technical literature, INSYDE-content leverages desk-based data to characterize model features, including uncertainty treatment arising from incomplete input data. A validation test on two historical flood events and a sensitivity analysis are performed to assess the model's performance and explore the contribution of input variables to damage estimation, confirming its robustness and interpretability. For illustrative purposes, in this study INSYDE-content has been tailored to the specific hazard, vulnerability and exposure characteristics of Northern Italy; nonetheless, its adaptable structure supports broader applicability across diverse regional settings, provided suitable customization is applied.

#### 1 Introduction

Floods rank among the most devastating disasters worldwide, with climate change projected to intensify both their frequency and severity through altered weather patterns (Brázdil et al., 2006; Hirabayashi et al., 2013; Tanoue et al., 2016; IPCC, 2023). In response to these growing threats, several flood management approaches have been formulated and implemented over the years to mitigate related losses, with a shift from focusing solely on flood hazard control to a more holistic flood risk management (Plate, 2002; Sayers et al., 2002; Gralepois et al., 2016; Disse et al., 2020).

In this change, damage estimation has become central for supporting the definition of effective policies for flood risk mitigation in both rural and urban environments (Merz et al., 2010; Marín-García, 2023). Modeling tools developed for this purpose vary in complexity, ranging from basic univariable formulations to more advanced multi-variable approaches involving several damage explicative factors. A further classification distinguishes between empirical and synthetic models based on the adopted methodology for their development: empirical models use historical data to relate vulnerability and hazard variables to damage, while synthetic models are founded on expert judgment for the definition of "what-if" scenarios supporting damage assessment (Merz et al., 2010; Jongman et al., 2012; Dottori et al., 2016; Gerl et al., 2016; Martínez-Gomariz et al., 2021). Synthetic and

multi-variable models offer valuable advantages, with the synthetic approach allowing for greater spatial and temporal transferability and the multi-variable formulation improving damage estimation reliability (Merz et al., 2010; Schröter et al., 2014; Wagenaar et al., 2018; Amadio et al., 2019; Scorzini et al., 2021; Paulik et al., 2023; Xing et al., 2023; Di Bacco et al., 2024).

Among the various exposed assets, damage modeling for the residential sector has been the most extensively investigated and developed, with numerous models available in the literature (Gerl et al., 2016). Nevertheless, most of these models are devised to assess building damage only, with household contents (i.e., those items within the house that are not permanently installed in it, such as the furniture and appliances) often overlooked or modeled through simplified approaches.

One of the early attempts to address the inherent complexity of appraising flood damage to house contents – stemming from the multiple economic, social and structural factors involved – is represented by the "Catalog of Residential Depth-Damage Functions" by the US Army Corps of Engineers (Davis and Skaggs, 1992). Through the inventory and valuation of household contents in sample houses, the report estimated that content damage may account for up to 50% of total building damage, thereby emphasizing its critical role in overall damage figures in inundation scenarios. Only more recently, a few studies have explored a variety of methodological approaches to improve the modeling capabilities for this asset across various spatial scales.

Zhai et al. (2005) analyzed flood damages for the 2000 Tokai flood in Japan, identifying inundation depth, duration and household income as key explicative variables of flood damage to both buildings and contents. Similar results were found by Wahab and Tiong (2016), who proposed a multivariate flood damage model for the residential sector in Jakarta, Indonesia. Romali and Yusop (2020) developed an empirical multiple regression flood damage model for Kuantan, Malaysia, incorporating residents' occupation, household income and building type as explanatory variables. They found that content loss was positively correlated with inundation duration along with household income and inundation depth. Shrestha et al. (2021) used household surveys in Myanmar's Bago region to create local flood damage functions based on building characteristics. Ahadzie et al. (2022) studied flood impacts on residential buildings in urban settlements across five flood-prone regions in southern Ghana, with data showing significant damage to contents, such as furniture and clothing.

Mosimann et al. (2018) developed and cross-validated two linear regression models based on insurance data to estimate building and content losses in Switzerland. Their findings revealed that contents accounted for 21% to 36% of total losses and even surpassed building losses in cases of low-severity damages. Endendijk et al. (2023) developed empirical, multivariate vulnerability models to assess flood damage to buildings and household contents in the Netherlands, incorporating the influence of flood mitigation measures on damage outcomes.

Nofal et al. (2020) addressed the issue of data scarcity and the challenges of deterministic models by developing synthetic uniand multi-variable, component-based, flood fragility functions. The method involved dividing the building into 65 components, with some of them representing household contents, such as chairs, desks, TVs, electric appliances, sofas, kitchen cabinets, beds and mattresses. Each component was assessed across five predefined damage states ranging from DS0 (insignificant damage) to DS4 (complete damage). The proposed approach was applied to a hypothetical single-family residential wood building, with contents distributed in it based on practical considerations. However, these basic assumptions significantly limit the model's real-world applicability, as they fail to account for the substantial variability in housing types and household contents.

Carisi et al. (2018) highlighted the challenges of developing comprehensive flood damage models for buildings and contents in the Italian context. Using ex-post data for the 2014 Secchia flood in the Emilia-Romagna region, they created empirical uniand multi-variable flood damage models for buildings. Damage to contents was instead assessed indirectly using a simple square root regression relationship to building damage. By overlooking the complex induction mechanisms leading to damage and the related key factors, such as content distribution within the building, this approach, although practical, significantly hinders the model's ability to provide insights into content vulnerability in flood scenarios.

Overall, the existing literature underscores the substantial complexity of modeling flood damage to contents, primarily due to the inherent variability of exposed items within the buildings and the difficulty in characterizing their susceptibility to flood. This variability tends to increase across regions, further complicating models' spatial transferability. Nonetheless, current research in this field remains limited, highlighting the need for further development of suitable methodologies and tools.

To address these gaps, this study presents INSYDE-content: a micro-scale, multi-variable synthetic model specifically developed to estimate flood damage for household contents. Building upon the traditional "what-if" approach and earlier expert-based efforts (e.g., Davis and Skaggs, 1992; Dottori et al., 2016), INSYDE-content combines probabilistic modeling with flexible input data requirements and explicit component-wise representation of damage mechanisms. This transparency and adaptability distinguish it from empirical models, which are often fitted to specific case studies, or from other synthetic models that operate as black boxes without allowing inspection or modification of internal assumptions.

The paper is structured around a four-step process, as depicted in Figure 1. The methodology section focuses on model development, starting with a preliminary data collection and analysis which support the identification and characterization of household content items, along with the formulation of the corresponding damage mechanisms. The procedure is illustrated for the context of Northern Italy (Po River District, Figure S1 of the Supplement Material 1), but the overall framework is generalizable and can be adapted to different regions. The role of input variables in shaping estimated content losses is then investigated through a probabilistic sensitivity analysis, with model performance ultimately tested against observed data from two historical flood events in Northern Italy.

## 2 Methodology

85

# 2.1 Model development

INSYDE-content adopts the general model framework proposed in the original INSYDE for buildings (Dottori et al., 2016), where damages are first modelled component-wise in physical terms and then converted into monetary values using the full replacement costs derived from reference price lists.

Figure 1: Overview of the workflow for the development and testing of INSYDE-content.

The overall economic damage D to house contents in each building is calculated by summing the replacement costs  $C_i$  for each of the n content items within the building that is expected to be damaged:

$$D = \sum_{i=1}^{n} C_i = \sum_{i=1}^{n} no.dam.items_i \cdot unitprice_i$$

where *no.dam.items<sub>i</sub>* represents the number of damaged items for each specific household content *i* and *unitprice<sub>i</sub>* the corresponding unit replacement prices. The value of *no.dam.items<sub>i</sub>* depends on flood event features (such as inundation depth and duration, flow velocity, and water quality, in terms of the presence of pollutants or sediments), as well as on the vulnerability and exposure of the affected objects:

$$no. dam. items_i = f(event features, content features)$$

INSYDE-content includes 11 standard items: beds, sofas, wardrobes, dining table setup (dining table and chairs), kitchen setup (lower and upper cabinets), TVs, washing machine, oven and microwave oven, dishwasher and refrigerator. Their selection was informed by an extensive review of real estate listings with photographic documentation, which provided a representative overview of typical household content arrangements. The goal was to identify a set of essential and commonly present items across different dwelling types, ensuring both relevance and generalizability of the model, thus supporting its applicability to a wide range of residential settings, while maintaining a manageable level of detail for damage estimation.

Since information on the actual number and distribution of contents within buildings is typically not known at large spatial scales, in model development it is necessary to introduce a probabilistic method that estimates their presence based on more commonly available data, such as building features. This approach enables a more accurate assessment of exposed items

without needing for detailed, point-wise evaluations, which are often impractical due to the highly subjective nature of individual household content choices. To this aim, in the tailored version of INSYDE-content for Northern Italy (Po River District), empirical data derived from virtual surveys of buildings have been used to establish reliable estimates of content distribution as a function of certain building features. The virtual surveys involve analyzing real estate listings to extract key information from advertised posts, architectural drawings and photos detailing the buildings and their contents (Scorzini et al., 2022). Given the potential inconsistency in data completeness and quality across real estate platforms, only listings with complete information about the building and its contents have to be considered. To qualify for selection, the advertisement has to include at least information on interior details, main geometrical attributes and architectural layout and profiles, as well as a sufficient number of photos taken from various angles to cover most of the areas within all rooms. The applied criteria in the analyzed case of the Po River resulted in only 60 houses being deemed suitable for analysis out of approximately 500 examined. Indeed, in many cases, only partial information could be obtained due to either a lack of data or privacy issues. Additionally, geometrical features did not always align with secondary data, such as building footprints available from other databases.

The information collected during this process includes both the characteristics of the building (e.g., location, number of floors, building type, finishing level, inter-story height, footprint area, surface area, external perimeter, year of construction) and those of its contents (e.g., number and size of beds and sofas, number of furniture pieces, placement height of appliances). For detailed examples, please refer to the Supplementary Material 2 included in this study.

For practical reasons, the raw data acquired from the survey need to be transformed into more standardized variables on a component-by-component basis. For instance, different types of beds (i.e., single and double) have been represented by a single standard variable called *BEDLeq*, where 1 *BEDLeq* equals 1 large double bed or 2 single beds. The same approach was applied to other items, such as sofas, wardrobes and dining table setup (for details, please refer to the Supplementary Material 1).

At this stage, it becomes possible to identify empirical relationships allowing for the assessment of the number of exposed contents per building by using regression functions that relate the standardized variables for each item to the characteristics of the building. A straightforward approach might involve correlating the number of specific items to the number of rooms designated for them (e.g., beds to the bedrooms). However, since the number of rooms per building is often an unknown information in flood risk assessments, it is essential to establish a relationship between the number of exposed items and more general independent variables to enhance the practical applicability of the model. The most straightforward approach is to relate item quantities to building size, expressed as either footprint area (FA) or surface area (SA). Footprint area refers to the total ground area occupied by the building, while surface area is the total horizontal built-up area across all floors. For single-story buildings, FA and SA are the same. For multi-story buildings, if each floor is occupied by separate households, the SA for each unit corresponds to the individual floor area. In cases where a single household occupies multiple floors, SA becomes more representative of the total contents, as it comprises the total floor area of the entire building (SA = NF·FA, with NF indicating the number of floors).

For Northern Italy, a power regression function of SA was found effective in describing the number of exposed beds, sofas, wardrobes and dining table setup (Supplementary Material 1). For these elements, a stochastic component derived from a normal distribution with a mean of 1 and a standard deviation of 0.2 was applied to account for the inherent variability in the distribution of household contents. The other items were instead treated as constant functions based on practical judgment supported by empirical observations: for instance, it is expected that each housing unit may contain only one kitchen setup.

The functions thus obtained can be then converted into step functions to more realistically represent the increments in the number of exposed objects with changes in SA. This means that for a specific range of SA, the standardized variable related to a certain content can assume only a physically sound constant value (e.g., the predicted *BedLeq* is expressed in increments of 0.5 *BedLeq* to capture the increase in the number of beds corresponding to a single bed unit).

Once the number of exposed items ( $exp_i$ ) is defined, it is possible to determine  $no.dam.items_i$  through the identification of the primary driving factors for damage induction for each content type. To this aim, INSYDE-content adopts a probabilistic approach based on the use of fragility functions. The model assumes a binary damage state: an undamaged state ( $ds_0$ ) or a fully damaged state ( $ds_1$ ). For each content type, fragility functions express the probability of reaching a fully damaged state, based on the event intensity measure(s) (IM). To combine this probability with the actual occurrence of a damage state for the individual exposed elements, a random value  $P_i$ , accounting for the survival probability of each item, is sampled from a uniform distribution between 0 and 1 and compared to the damage probability derived from the fragility function for the corresponding content type. The random nature of the implemented process serves to capture the inherent uncertainty in the damage mechanisms, reflecting the intrinsic variability in content vulnerability to the same event intensity. Consequently, if  $P_i$  falls below the damage probability calculated from the fragility function, it is considered fully damaged ( $ds_1$ ), otherwise, it remains undamaged ( $ds_0$ ).

The formulation of the fragility functions is based on expert knowledge, practical experience, as well as available technical and scientific documentation. Assigning the thresholds for the event feature driving the damage mechanism is not trivial and the lack of relevant research in this field adds further complexity to it. However, given the practical considerations surrounding the damage process for contents, it is feasible to establish lower and upper bounds for the IM based on expert judgment.

In the model, inundation depth and duration are identified as the primary drivers of damage to contents, while floodwater quality (i.e., the presence of sediments or pollutants) is considered a secondary stressor, whose influence is accounted for through a scaling factor that amplifies the damage probability. As an illustrative example, the case of beds is presented here, while information for the other components is provided in the Supplementary Material 1. Fragility curves for both inundation depth and duration are modeled using truncated normal distributions, bounded by thresholds that are physically meaningful. Specifically, the thresholds for depth range from a few centimeters above the floor level to the typical base height of a mattress, while even short durations of inundation (on the order of a few hours) may be sufficient to cause damage due to water absorption. The model evaluates the probability of damage for each primary factor independently and assigns the maximum of the two based on the assumption that the most unfavorable condition dominates the damage mechanism, independently of

the other. The presence of sediments in the floodwater is instead assumed to increase the probability estimated from the main primary factor by 10%, reflecting the higher likelihood of irreversible damage due to debris or fouling.

Tables 1 and 2 describe the building and event features used in the model, including their range of values and the assumptions regarding the dependencies between the variables in case of missing information. Indeed, similar to the original INSYDE for buildings, the model can automatically assign default values when certain input data are not provided by the user. This is achieved by either leveraging the implemented relationships among the variables (e.g., for SA or h<sub>i</sub>) or by sampling from user-defined distributions that reflect the characteristics of the region under analysis.

Table 1: Building features in INSYDE-content for estimating exposed household items.

| Variable Description |                                  | Range of values                                                                    | Default dependencies |  |
|----------------------|----------------------------------|------------------------------------------------------------------------------------|----------------------|--|
| NF                   | Number of floors [-]             | > 0                                                                                |                      |  |
| FA                   | Footprint area [m <sup>2</sup> ] | > 0                                                                                |                      |  |
| SA                   | Surface area [m²]                | > 0 SA=f(FA, NF)                                                                   |                      |  |
| ΙΗ                   | Inter-floor height [m]           | > 0                                                                                |                      |  |
| GL                   | Ground floor level [m]           | $\geq 0$                                                                           |                      |  |
| BT                   | Building typology [-]            | <ol> <li>Detached House</li> <li>Semi-Detached House</li> <li>Apartment</li> </ol> |                      |  |
| FL                   | Finishing level [-]              | 0.8: Low<br>1: Medium<br>1.2: High                                                 |                      |  |
| GU                   | Ground use [-]                   | 1: Residential use<br>2: Other use (garage, storage, etc.)                         | GU=f(BT)             |  |
| HU                   | Number of housing units [-]      | ≥ 1                                                                                | HU=f(FA,BT)          |  |

Table 2: Event features in INSYDE-content.

| Variable | Description                                               | Range of values | Default dependencies |
|----------|-----------------------------------------------------------|-----------------|----------------------|
| he       | Inundation depth outside the building [m]                 | $\geq 0$        |                      |
| hi       | Inundation depth inside the building (for each floor) [m] | [0:IH]          | hi=f(he,GL)          |
| d        | Inundation duration [hours]                               | $\geq 0$        | d=f(he)              |
| S        | Indicator for the presence of sediments [-]               | 0: No           |                      |
|          |                                                           | 1:Yes           |                      |
| q        | Indicator for the massacra of mallytants []               | 0: No           |                      |
|          | Indicator for the presence of pollutants [-]              | 1:Yes           |                      |

In the version presented in this study, the model incorporates the distributions proposed by Di Bacco et al. (2024), which are based on a combination of physically-informed approaches and empirical survey data for northern Italy (reported for completeness in Figures S2-S3 of the Supplementary Material 1).

Additionally, for a more comprehensive analysis, this study also employed the non-region-specific synthetic distributions proposed by Di Bacco et al. (2024) (Figures S4-S5 in Supplementary Material 1). By covering a wider range of values, these

distributions facilitate a deeper investigation into the model's sensitivity to input variables beyond the specific context of northern Italy. The fragility functions (FF) and general assumptions for all components are detailed in the Supplementary Material 1, while Table 3 summarizes the variables that affect both the number of exposed elements (EXP) and the damage mechanisms in the model. As evident from the table, EXP is not only expressed as a function of the floor area, but also depends on building characteristics (e.g., typology, number of floors and ground floor use), as the internal distribution of contents is shaped by the structural and functional layout of the dwelling, which may vary considerably between apartment and single-family buildings. For instance, in detached houses, bedrooms are typically located on upper floors, affecting the vertical distribution of certain elements such as beds and wardrobes.

It is worth noting that flow velocity does not appear among the event features considered in INSYDE-content (Table 2), despite its potential relevance to content damage. The decision to exclude it as an input variable was guided by practical considerations. Firstly, the impact of flow velocity on content damage should ideally be assessed based on water velocity inside the building. However, this information is neither available in standard flood risk analyses nor are there reliable methods to estimate it indirectly from external flow velocities (Dewals et al., 2023; Zhu et al., 2023). Furthermore, the mechanism of flow intrusion into a building generally leads to a significant reduction in velocity, making its effect negligible when assessing content damage. Conversely, when the external flow velocity is extremely high and capable of washing away the entire building, flow velocity may become critical as it would lead to the complete destruction of both the structure and its contents. However, this extreme scenario has not been included in the current version of INSYDE-content due to its rarity in fluvial floods, particularly for masonry and reinforced concrete buildings. Additionally, the lack of well-established criteria defining the threshold for washing away (based on a combination of water depth and velocity) makes it challenging to quantify a situation where content damage would be complete.

Table 3: Variables that affect the determination of exposed elements (EXP) and the damage mechanisms (i.e., fragility functions (FF)) in INSYDE-content.

| Damage component                        | Variables affecting the component         |  |  |
|-----------------------------------------|-------------------------------------------|--|--|
| Beds, wardrobes, dining setup           | EXP: SA=f(FA,NF), BT, HU=f(FA,BT), NF, GU |  |  |
| Beds, wardrobes, diffing setup          | FF: hi=f(he, GL), d, s, q                 |  |  |
| Sofas                                   | EXP: SA, BT, FL, HU, NF, GU               |  |  |
| Solas                                   | <i>FF</i> : hi, d, s, q                   |  |  |
| Kitchen setup                           | EXP: HU, NF, GU                           |  |  |
| Kitchen Setup                           | <i>FF</i> : hi, d, s, q                   |  |  |
| Washing machine, TV, oven, refrigerator | EXP: HU, NF, GU                           |  |  |
| washing machine, 1 v, oven, renigerator | <i>FF</i> : hi                            |  |  |
| Dishwasher, microwave oven              | EXP: HU, FL, NF, GU                       |  |  |
| Distiwastici, iniciowave oven           | FF: hi                                    |  |  |

## 2.2 Model evaluation

225

230

245

250

## 2.2.1 Sensitivity analysis to the input variables

A sensitivity analysis of INSYDE-content was conducted to evaluate how the selected explanatory variables influence damage estimation and to determine how the absence of certain input data contributes to uncertainty in these estimates. This analysis was performed using two synthetic portfolios, each consisting of 250,000 buildings exposed to hypothetical flooding scenarios, as described in Di Bacco et al. (2024). The first dataset represents more general, non-region-specific inundation and building characteristics (referred to as the "extended dataset", hereinafter), while the second focuses on the specificities of northern Italy (Po River District, "Po dataset", hereinafter). These synthetic portfolios were generated by Di Bacco et al. (2024) by integrating empirical distributions of both hazard and exposure variables, while preserving their mutual dependencies and accounting for internal variability (Figures S2-S5 in Supplementary Material 1). The Po dataset is based on a combination of official inventories and virtual survey data representative of the local building stock and flood dynamics, whereas the extended dataset spans a wider range of conditions to support more generalizable insights. This dual approach allowed us to derive a context-specific and a broader ranking of variable importance, and to investigate how regional characteristics may influence the relative weight of each feature within the model.

The sensitivity analysis followed these steps: first, INSYDE-content was applied to calculate damages for both building portfolios, with all required model variables known. For each j-th building in the dataset, the estimated damage value in this step was taken as reference value, D<sub>0</sub>. Then, one input variable was sequentially removed and replaced, for each building in the datasets, with values sampled from the distributions given by Di Bacco et al. (2024). This process was repeated for each variable, with damage recalculated each time (D<sub>i</sub>). The absolute difference in damage compared to the reference value was recorded for each of the j buildings (| D<sub>0</sub> – D<sub>i</sub> |<sub>j</sub>), facilitating the determination of the variance each feature contributes to the model's outcome.

#### 2.2.2 Model validation

The model was validated using loss data from two historical flood events that occurred in Northern Italy: the 2002 flood in Lodi and the 2010 flood in Caldogno. These events have been analyzed in previous studies regarding building damage (Amadio et al., 2019; Molinari et al., 2020), but never for household contents. In this study, only buildings without basements were considered, coherently with the model's assumption that does not account for their presence. The validation dataset includes 194 buildings for Lodi and 169 buildings for Caldogno, with total losses (adjusted to year 2023 values) amounting to about 3.1 million euro for each case.

The loss data were derived from the forms for damage quantification distributed by the authorities as part of the state's loss compensation process, which were filled in by affected citizens. These forms provided actual restoration costs, certified by original receipts and invoices. The types of content damage eligible for compensation aligned with the 11 components identified in the model formulation. The unit prices for content items were taken from the decree issued by the delegated

commissioner responsible for damage compensation following the Caldogno flood. These prices were then applied to the Lodi case after being adjusted for inflation to the event date.

In addition to loss data, the dataset contains information on external water depth at building's location. As regards the other event features required by INSYDE-content, qualitative data from previous studies allowed to identify an approximate value for inundation duration equal to one day for both cases (Di Bacco et al., 2024). Accordingly, the sampling of d values was obtained from a truncated normal distribution centred at 24 hours and spanning between 16 and 48 hours (d\* in Table 4). The information on the presence of fine-graded sediments allowed for assigning s=1 (yes) for both cases, while local data on pollutants was available only for Lodi.

Concerning building features, the dataset includes footprint area (FA), number of floors (NF), building type (BT) and finishing level (FL). For the remaining missing variables (hi, SA, GL, GU, IH, HU) the model was applied by leveraging the established relationships among the variables (Tables 1 and 2) and the built-in sampling process from pre-defined distributions, as outlined in Di Bacco et al. (2024).

To account for uncertainty arising from the sampling process of unknown inputs and from the implemented assumptions on exposure and damage mechanisms, content losses were calculated probabilistically for each building over 1,000 iterations, resulting in confidence intervals for the estimated values which were compared to reported losses.

Table 4: Available input variables for INSYDE-content in Caldogno and Lodi case studies.

| Case study | Variables included in the dataset |
|------------|-----------------------------------|
| Caldogno   | he, d*, s, FA, NF, BT, FL         |
| Lodi       | he, d*, s, q, FA, NF, BT, FL      |

#### 3 Results and discussion

## 270 3.1 Examples of resulting damage functions

Figure 2 illustrates an example of traditional damage functions resulting from the application of INSYDE-content, broken down by components and in terms of total damage. The functions correspond to a scenario involving both apartment and single-family, detached or semi-detached residential buildings with a surface area of 200 m<sup>2</sup>, distributed across two floors, with IH=3 m, GL=0 m and a high finishing level. In the total damage functions, the interquartile range is shown alongside the median damage to represent the variability induced by the probabilistic modeling of the damage mechanism. For each water depth, damage values were calculated over 1,000 iterations by randomly assigning q as either 0 or 1 and generating random samples of inundation durations based on water depth scenarios, with d assumed to be relatively short for shallow inundations and extending from 6 hours to 2 days for depths greater than 2 meters.

The trends shown in Figure 2 effectively highlight the assumptions on the damage mechanisms implemented into the model for the various items, as described in the Supplementary Material 1. For instance, focusing on detached and semi-detached

buildings, damage to beds only occurs when inundation depth exceeds 3 meters, consistent with the assumption that bedrooms are located on the upper floors for this type of dwelling. Similarly, sofas, assumed to be in the downstairs living area, appear to be highly vulnerable to flood, necessitating a full replacement even under very shallow water depths. The pattern for wardrobes and their contents is more complex, as they include different types of furniture, ranging from small and decorative pieces in the living room to full-height wardrobes in bedrooms, each with varying thresholds for damage onset.

Figure 2: Example of traditional damage functions resulting from the application of INSYDE-content for different building types: a) total content damage; b) damage to kitchen and damage setup as well as electric appliances; c) damage to beds, sofas and wardrobes to a detached/semi-detached building; d) as panel c), but for an apartment building.

This results in the stepped function shown in Figure 2, panel c), similar to the one for the kitchen setup, where the steps represent the damage occurring to lower and upper cabinets. Electrical components display a similar behavior across all items, with only different activation thresholds based on their varying vulnerability. For apartments, the overall damage is higher compared to single-family buildings with the same surface area, due to the presence of multiple housing units within the same space, each containing its own furniture and appliances. To ensure clarity of Figure 2, panel b) focuses exclusively on the damage functions for electrical components and kitchen setup in a single-family building, which are equivalent to those for a single housing unit in an apartment building. For the other components, distinct patterns emerge between the two building types, reflecting different assumptions regarding the distribution of contents across the floors. For example, in the case of apartments, a non-null damage to beds can be recognized even on the first floor, with a stepped increase in damage between floors, as observed for sofas.

## 3.2 Sensitivity analysis to the input variables

This section reports on the sensitivity of damage estimation to the variables considered in INSYDE-content and on the uncertainty arising from potential missing input data in model implementation. Figure 3 summarizes the findings by illustrating the difference in computed damage when the model is applied to the reference portfolios of 250,000 buildings and to their replicas obtained by replacing the values of one input variable at a time with a sampling from the predefined distributions implemented in the model.

Figure 3: Variable importance in INSYDE-content. Results obtained with sampling from the two different datasets developed by Di Bacco et al. (2024): a) case for the extended synthetic dataset; b) case for the Po River District synthetic dataset. Variables are ordered according to the median value of the absolute damage difference.

The left and right panels of Figure 3 correspond, respectively, to the extended synthetic dataset, which encompasses a broader range of values for the input features, and the Po dataset, tailored to the characteristics of northern Italy. In Figure 3, features are ranked by the median value of  $|D_0 - D_i|_j$ , where  $D_0$  represents the reference damage for the j-th building and  $D_i$  the corresponding value calculated after removing and resampling the i-th variable from the predefined distributions developed by Di Bacco et al. (2024).

Dependent variables, such as SA or hi, are not represented in the figure as their effects are inherently incorporated through the independent variables to which they are assumed to be functionally related. Similarly, the influence of sediments, represented by the binary variable s, is not reflected in the figure; this is because the distributions generated by Di Bacco et al. (2024) for representing the fraction of sediments on water volume always yield a non-null value in the analysis, which implies a constant s=1 (i.e., presence of sediment, of any gradation) in INSYDE-content.

Overall, Figure 3 demonstrates a feature importance pattern that remains relatively consistent across the two tested datasets, with the most influential variables (he, BT, FA, and d) consistently appearing among the top five in both cases. As expected, higher values of  $|D_0 - D_i|_j$  are observed in the extended dataset, where missing data sampling (especially for extensive and intensive variables) can span a broader range of variability than in the Po scenario (Di Bacco et al., 2024).

Inundation depth (he) emerges as the primary contributor to uncertainty in damage estimation, with median absolute damage differences ranging from approximately 4,500 to 6,800 euro, confirming the well-known significance of such variable in direct flood damage assessment for various types of assets, including household contents (Merz et al., 2013; Schröter et al., 2014).

Building type (BT) plays a crucial role in determining the exposed value of contents by shaping the distribution of items within the building (e.g., the assumption regarding bed placement in single- and multi-family buildings discussed earlier). This effect is evident in the median  $|D_0 - D_i|_j$  shown in Figure 3, which range from approximately 3,000 to 3,500 euro, establishing BT as the second most influential feature in INSYDE-content, in contrast to its minimal impact on building damage mechanisms (Dottori et al., 2016; Di Bacco et al., 2024). The minor influence of ground use (GU) is partly masked by the effect of BT, as the model differentiates the use of the ground floor only in the case of apartment buildings, where it is more common for the

Closely related to inundation depth is the role of ground level (GL), which influences the water depth inside the building and subsequently affects damage. In the extended dataset, GL ranks as the third most important feature, with median absolute damage differences of about 2,900 euro. On the other hand, in the Po dataset, the lower variability of GL makes its impact less significant, resulting in an estimated damage variation of about 1,500 euro, which is comparable to variations observed for other variables, such as FL, GU and q.

ground floor to be allocated for non-residential purposes (e.g., garages, communal areas, etc.).

Regarding the number of floors (NF), while the quantitative effects on damage calculations are similar in both datasets, with a median  $|D_0 - D_i|_j$  of approximately 1,700 – 2,000 euro, greater variability is observed in the extended dataset (with an interquartile range of 5,400 euro, compared to 3,100 euro for the Po case), as a consequence of the higher probability of damage affecting upper floors, given the broader range of inundation depths represented in this dataset.

For the Po case, a relatively stronger influence is observed for the two variables affecting the number of exposed contents - either directly through footprint area (FA) or indirectly via finishing level (FL) (see Supplementary Material 1 for the corresponding functions) - with these variables occupying the third and fifth positions, respectively. The relatively limited impact of FL, as shown in Figure 3, can be attributed to the fact that the model considers its effect only for the estimation of the number of exposed contents, without accounting for potential increases in their unit price. This modeling choice was guided by practical considerations: first, for household contents, it is virtually impossible to establish a clear upper limit on the economic value of an item as its luxury level increases; second, this approach aligns with the application of the model in flood risk management contexts (e.g., under the European Floods Directive) or in government compensation schemes for flood damage, where standard average costs are used, regardless of the actual quality of the damaged items.

## 3.3 Insights into content-to-building damage relationship

This section expands on the model's performance by analyzing the relationship between content and building damage, aligning with previous studies that aimed to establish connections between the two (Thieken et al., 2005; Carisi et al., 2018; Mosimann et al., 2018). To this end, building damage was calculated using the INSYDE 2.0 model (Di Bacco et al., 2024) and paired with content damage estimates for the two synthetic datasets used in the sensitivity analysis. Figure 4a illustrates the results, plotting estimated content damage against building damage for both datasets and comparing them with the equation proposed by Carisi et al. (2018), based on post-event observations for the 2014 Secchia flood in Emilia Romagna, Italy.

Figure 4: a) Content to building damage calculated with INSYDE models on the two tested datasets and comparison with the root function of Carisi et al. (2018); b) Content to building damage ratio (CBR) expressed as a function of inundation depth with INSYDE models and Carisi et al. (2018).

INSYDE results reveal substantial variability in content damage for same levels of building damage, highlighting the complex, multi-variable nature of damage mechanisms that cannot be fully represented by simple univariate functions. Interestingly, the root function proposed by Carisi et al. (2018) aligns at a median level with INSYDE results, although in a region with lower sample density. Conversely, INSYDE allows discerning distinct patterns in the relationship between building and content damage, shaped by the combined effects of inundation duration thresholds and building characteristics (such as BT and FL) triggering specific damage mechanisms to certain components, as previously described for INSYDE 2.0 by Di Bacco et al. (2024).

Furthermore, Figure 4b illustrates the content-to-building damage ratio (CBR) as a function of inundation depth, offering insights into the limitations of using a univariate approach based solely on building damage, especially in case of shallow inundation depths (< 0.5 m), where the higher vulnerability of contents can lead to CBR values exceeding 5. However, as inundation depth increases, overall damage becomes predominantly driven by building damage, leading to a reduction in the differences between the two approaches.

Median CBR values for INSYDE across the two datasets range from 0.26 (calculated on inundation durations exceeding 48 hours) to 0.36 and 0.42, respectively for the extended and Po River datasets at shorter durations. These findings are consistent with post-event observations in Switzerland and Germany (Thieken et al., 2005; Mosimann et al., 2018), which reported CBR values around 0.28-0.29, with higher values for lower-entity damages, as illustrated here in Figure 4b. Nonetheless, the results presented in Figure 4 highlight the importance of detailed knowledge regarding the vulnerability and exposure characteristics of potentially impacted assets to achieve accurate content damage estimations with minimized uncertainty. The probabilistic sampling approach implemented in INSYDE-content addresses the practical challenges of obtaining such detailed information in large-scale applications by effectively managing unknown missing data through calibrated distributions of the input features, while also explicitly accounting for uncertainties associated with predictions and thereby mitigating the false sense of certainty that, instead, often accompanies deterministic models.

#### 3.4 Model validation

390 The validation outcomes are summarized in Table 5, which presents the statistics of total damage estimated by INSYDEcontent for the two considered historical case studies, based on 1,000 replicates for each building with missing input features, alongside observed damage values. Figure 5 complements these findings with a detailed visualization of the differences between individual building-level estimations and actual observations.

Overall, Table 5 indicates a good alignment between the estimated and observed total damages. In Lodi, the sum of median predicted losses closely reflects the actual claim data, while in Caldogno the reported figure approaches the upper quartile of the estimate distribution. Despite this general convergence at the aggregated level, Figure 5 reveals notable scatter at the individual building scale, particularly for apartment buildings, where the absence of ground-level usage information (GU) introduces large variability in the estimates, as the model randomly samples residential or non-residential use over the iterations, affecting both the exposed contents and the resulting damage outcomes. In contrast, predictions for detached and

semi-detached houses show much lower variability, likely due to their more homogeneous features and to the relatively uniform, shallow inundation depths recorded in the two flood events (approximately 0.5 m in Caldogno and 0.8 m in Lodi, on average).

This scatter is also evident in quantitative error metrics computed between the median predicted values per building and the individual observed losses: the mean absolute error ranges from 14,185 euro for Caldogno to 16,962 euro for Lodi, with corresponding mean biases of -8,239 euro and 1,233 euro. Nevertheless, while informative, these building-level analyses should be interpreted with caution for two main reasons. First, flood damage models are widely acknowledged in the literature to perform better at more aggregated spatial scales (e.g., municipality or regional scales), as this helps to compensate for building-specific variability and inconsistencies in the input data (e.g., Merz et al., 2008; Molinari et al., 2020; Pinelli et al., 2020). Moreover, such issues are particularly critical for content-related damages, which are inherently more variable and less standardized than building elements, making them especially challenging to model.

Table 5: Results of the probabilistic validation of INSYDE-content for the case studies of Caldogno and Lodi: statistics of total estimated damage versus reported damage.

|          | Estimated damage [M€ 2023] |        | Observed damage [M€ 2023] |                           |
|----------|----------------------------|--------|---------------------------|---------------------------|
| _        | 1st quartile               | Median | 3 <sup>rd</sup> quartile  | Observed damage [Me 2025] |
| Caldogno | 1.10                       | 1.76   | 3.50                      | 3.15                      |
| Lodi     | 1.75                       | 3.30   | 6.87                      | 3.06                      |

Figure 5. Results of the probabilistic validation of INSYDE-content: a) Caldogno event; b) Lodi event. Median computed damage (dot) and corresponding interquartile range (line) are plotted for each building against observed damage (expressed in 2023 euro).

Second, claim data themselves should not be treated as an unequivocal ground truth, as they are often affected by substantial uncertainties and potential biases (Molinari et al., 2020; Pinelli et al., 2020; Wing et al., 2020; Museru et al., 2024). Consequently, the concept of "validation" in flood damage modeling should be interpreted with caution and possibly reconsidered in light of these constraints (Molinari et al., 2020). In support of these considerations, Supplementary Figure S7 shows the relationship between observed content losses and inundation depth at the building level. This figure clearly illustrates the large, non-physical variability in the claim data, with differences in losses of up to an order of magnitude even for buildings of the same type and size affected by nearly identical flood conditions. For instance, two identical single-family houses located side by side and subject to approximately 30 cm of inundation recorded claim amounts of about 600 and 16,000 euro, respectively. Such discrepancies cannot be explained by physical processes alone, but rather reflect non-physical variability inherent in claim reporting practices (including differences in household behavior, repair and replacement choices or attitudes toward compensation mechanisms) that cannot be reproduced by physically informed models such as INSYDE-content.

Furthermore, in addition to confirming the model's robustness in estimating aggregated losses, the analysis also highlights the value of explicitly representing uncertainty through prediction intervals when assessing model performance, thereby enhancing transparency and helping to avoid the misleading impression of accuracy often conveyed by deterministic approaches (Pappenberger and Beven, 2006; Merz et al., 2015).

#### 4 Conclusions

Modeling flood damage to household contents poses significant challenges due to the inherent heterogeneity of such items, which contrasts to the more standardized geometry and features of buildings. This complexity is further exacerbated by the need to account for a wide range of factors influencing damage mechanisms (i.e., flood characteristics, the vulnerability of both buildings and their contents, as well as broader economic aspects), demanding a more comprehensive approach than traditional univariate damage assessments.

This study aimed to address such complexities by introducing INSYDE-content, a probabilistic, multi-variable flood damage model, specifically designed for household contents. The model was developed through a structured, multi-phase process including: (i) an extensive data collection on household contents and flood characteristics; (i) the probabilistic estimation of the presence of household contents to reflect their variability and distribution across different building types; (iii) the development of fragility functions for individual items based on their driving factors; and (iv) a price analysis to obtain monetary damage estimations. Although the model presented in the paper has been tailored to the context of Northern Italy (Po river district), its components and basic input data can be adapted or adjusted to the specific characteristics of other regions, ensuring its generalizability.

Sensitivity analysis and validation exercises provided insights into the model's performance. One of the key findings emerging from this study is the critical importance of accounting for the multi-variable nature of flood damage to household contents, with the primary factors being inundation depth (and, to a lesser extent, inundation duration) and the variables that quantify

and distribute exposed items within buildings, such as those representing building size, type and use. The study also confirms that in cases of shallow inundation, content damage can exceed building damage, with content-to-building damage ratios largely exceeding 1, thus highlighting the often-overlooked vulnerability of household contents in flood scenarios.

The probabilistic nature of INSYDE-content effectively addresses uncertainties arising from potential missing input data, offering a more robust alternative to traditional deterministic models. By employing sampling techniques of input features, the model can estimate damage even in the absence of certain data, providing information on estimation uncertainty while maintaining transparency. Furthermore, the validation results confirmed the reliability of INSYDE-content, with its aggregated damage estimates aligning with observed claim data from two flood events in Italy. From another perspective, the analysis performed in this study also highlighted the importance of incorporating input data uncertainty when validating and evaluating model performance. Presenting damage estimations with clearly defined uncertainty bounds not only enhance model transparency and reliability, but also helps preventing a false sense of certainty, which is typical of simpler deterministic models. On the other hand, comparing observed data within a plausible range of the estimates can also help to support a more realistic appreciation of the uncertainty affecting the same claim data.

By modeling the complex interactions between flood characteristics, building features and household contents, while also addressing the challenges posed by data availability and uncertainty, INSYDE-content provides a robust and reliable tool for comprehensive flood risk assessment. Additionally, its modular architecture, offering a transparent representation of damage mechanisms at the component level, makes the model particularly suited not only for loss estimation, but also for the design and assessment of targeted mitigation or adaptation measures at the building scale.

Future work should focus on expanding and customizing the input datasets for household contents across different geographical contexts, refining and validating component fragility functions for a wider range of flood scenarios, and exploring the role of additional socio-economic factors in shaping damage patterns.

# Code availability

During the review process, the code of INSYDE-content is available at the following link:

https://drive.google.com/file/d/1T8XPF5UPALxJ0JGUYj2rodl7TStRP9Pz/view?usp=sharing and it will be made available on Mendeley data upon final acceptance.

#### 475 Author contribution

Pradeep Acharya: Data curation; Formal analysis; Investigation; Software; Visualization; Writing - review and editing.

Mario Di Bacco: Conceptualization; Methodology; Data curation; Investigation; Software; Writing - review and editing.

Daniela Molinari: Conceptualization; Investigation; Writing - review and editing.

Anna Rita Scorzini: Conceptualization; Methodology; Investigation; Visualization; Writing - original draft.

## 480 Competing interests

The authors declare that they have no conflict of interest.

## Financial support

This study was partly carried out within the RETURN Extended Partnership and received funding from the European Union Next-GenerationEU (National Recovery and Resilience Plan - NRRP, Mission 4, Component 2, Investment 1.3 - D.D. 1243 2/8/2022, PE0000005).

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
