# Peer review of "INSYDE-content: a synthetic, multi-variable flood damage model for household contents"

_EGUsphere, 2025_

## Author Comment (AC1)

**Reviewer #1**

**R1.C1**: *This paper provides a detailed description of the development of an expert based flood content damage model called INSYDE. It seems to be a follow up paper on the structure damage version of INSYDE, a model that seems to have found quite widespread use in the literature. The paper is well written and describes the development process well. The methods are solid but not very innovative and have been around in the grey literature for a long time (e.g. US Army Corps of Engineers). This paper goes in quite some detail describing the methods and adds much needed validation and is therefore definitely a useful addition to the scientific literature. That being said I have concerns about the validation results and more importantly the analysis of the results*.

**Reply**: We thank the Reviewer for the positive assessment of the manuscript. While we acknowledge the existence of various kind of damage models in the literature, we want to underline that INSYDE-content explicitly applies the synthetic approach to the context of content damage in residential buildings, a domain in which models are scarce. The proposed model was indeed designed following a component-wise, probabilistic and "what-if" logic, which builds on and extends the original INSYDE framework (Dottori et al., 2016). Its transparent and modular structure allows the users to clearly understand all assumptions and input variables, facilitating local adaptation based on available knowledge and data. In this respect, INSYDE-content overcomes limitations of both empirical models (typically fitted to specific case studies) and other synthetic models, which often function as black boxes without allowing modifications or explicit access to internal assumptions. While this aspect was briefly introduced in the original manuscript (L80-83), we will revise the paper to clarify and strengthen this point in the revised version.

**R1.C2**: *Figure 4 shows that for detached and semi-detached houses the variation in observed damages is much larger than the variation in predicted damages. My first impression is that the model always roughly predicts the same damage regardless of the circumstances (the blue dots are a nearly horizontal line). I think it may not be so bad because the log-log scale masks some of the variation. However, more information is required so readers can actually tell the model performance. For example, I currently cannot see if the variation in observed values is just based on some large outliers or whether there is some more fundamental problem whereby the observed losses have much more variation than the modelled losses. Also is there even any correlation between modelled and observed losses?*

**Reply**: We thank the Reviewer for the valuable comment. As also discussed in Molinari et al. (2020), claim data at the building scale are often affected by significant uncertainty and potential bias. For this reason, traditional flood damage models, as well as INSYDE-content, are typically more reliable when applied at aggregated spatial scales, rather than at the level of individual buildings.
This intrinsic variability in the observed data makes the interpretation of building-scale validation results particularly challenging. In our original manuscript (L368-374), we emphasized the importance of providing uncertainty ranges in the predicted damage values as a way to enhance the informative content of the model compared to purely deterministic approaches.
Nonetheless, we acknowledge the Reviewer's request for a clearer assessment of model performance. In the revised version of the manuscript, we will include standard error metrics (calculated based on the median of the predicted values per building, compared to the observed losses) to complement the visual inspection of the plots. However, we will also add a caveat to caution against over-interpreting these metrics, since observed values should not be considered absolute ground truth in damage modeling, due to the inherent limitations in claim data quality.
As also noted by the Reviewer, the apparent flatness of the predicted damage values for detached and semi-detached houses is partially explained by the use of log-log axes, which compress the visual perception of variability. Furthermore, from a theoretical perspective, a limited dispersion in predicted

losses is to be expected for these building types, given their relatively homogeneous characteristics and the shallow inundation depths recorded during the two flood events. In contrast, greater variability is observed for multi-family residential buildings, which generally exhibit broader heterogeneity in both exposure and vulnerability, leading to capturing a damage prediction variability of the same order of that for observed losses. In the revised version of the manuscript, we will therefore include comments on these aspects regarding the interpretation of the results shown in Figure 4.

**R1.C3**: *I understand that there is unexplained uncertainty in the model predictions as indicated by the uncertainty ranges in figure 4. However, if the model typically predicts more or less the same mean how do I know such a complicated model adds any value compared to a simple mean value as prediction?*

**Reply**: As also noted in our response to R1.C2, the apparent lack of variation in the predicted values is partly an effect of the log-log scale used in the plots, which visually compresses differences in magnitude. Moreover, for the two case studies presented, the variability in observed inundation depth was relatively limited, leading to a correspondingly limited spread in the predicted central estimates.
However, we believe that the added value of our approach lies in several aspects beyond the mean prediction. First, INSYDE-content is a probabilistic, component-based model that explicitly propagates uncertainty from input variables to output damage estimates. The resulting prediction intervals offer critical information to decision-makers, allowing them to understand the sensitivity of outcomes to input assumptions and to assess risk under uncertainty, an aspect that cannot be captured by a simple mean-based model.
Second, beyond estimating damage, the model provides a transparent and flexible structure to represent damage mechanisms explicitly. This feature allows, for instance, for scenario testing and the evaluation of building-scale mitigation strategies, which would alter specific input parameters and therefore result in different damage outcomes. This functionality makes INSYDE-content particularly suitable not only for risk estimation but also for supporting risk reduction planning.
In the revised version of the manuscript, we will better highlight these aspects, including a clearer explanation of how the probabilistic design and modular structure of the model enhance its applicability and usefulness beyond average damage prediction.

**R1.C4**: *Also very common error metrics are missing such as Mean Absolute Error, correlation coefficient or R2, so it's nearly impossible to assess how the model is doing from the information presented in the paper. Not all these metrics are needed but at least more information. Table 4 only gives an aggregated comparison, so basically gives a bias value. In one region there seems to be some bias but the authors do not really explain where this bias might be coming from. Lastly, I would expect an in depth analysis and discussion of the model performance in the paper based on the validation. That analysis is missing, making the validation not very useful in its current form.*

**Reply**: As discussed in our response to R1.C2, our original decision not to include commonly used error metrics was driven by the recognition that claim data at the building scale are themselves often affected by considerable uncertainty and potential bias. Consequently, they cannot be considered a definitive benchmark for validation purposes in the traditional sense. For flood damage models, quantitative validation against building-level claims must be indeed interpreted with caution, as it often reflects discrepancies not only in model performance but also in data quality and reporting practices (Molinari et al., 2020; Di Bacco et al., 2024). That said, we understand the Reviewer's request for a more detailed assessment. In the revised version of the manuscript, we will therefore include a set of standard error metrics, computed using the median predicted value per building, to provide a clearer quantitative basis for comparison with observed claims, and we will add a broader reflection in the discussion section on the limitations and interpretability of validation results in the context of flood damage modeling.

**R1.C5**: *Some of the input variables for the model validation seem sampled whereas others seem observed and the current text is very unclear about what is sampled and what is observed. This makes it even more difficult to interpreted the validation results.*

**Reply**: We agree that a clearer distinction between observed and sampled input variables can be useful for better understanding the setup of the validation exercises and interpreting the results. To address this, in the revised version of the manuscript we will include a summary table listing the model input variables used in the two validation case studies, clearly indicating for each variable whether it was directly observed or sampled from distributions.

**R1.C6**: *The word "to" in the title doesn't read well, maybe you can replace it with "for"? Or another solution.*

**Reply**: In the revised version of the manuscript, we will update the title to improve readability. The new title will be: "INSYDE-content: a synthetic, multi-variable flood damage model for household contents".

---

## Author Comment (AC2)

**Reviewer #2**

**R2.C1**: *The paper presents a new model for estimating flood damage to houshold contents. In contrast to damage to residential buildings, damaged household items are neglected in many (scientific) flood damage models or are estimated using simple approaches such as a lumped share of the estimated building damage. In practical loss estimation applications such as cost-benefit-analysis, where a loss estimation for all sectors and damage types is needed, further approaches exist, e.g. specific stage-damage functions for contents. These approaches are often not well documented or published. With INSYDE-contents the authors propose a detailed flood damage assessment of household contents based on 11 typical household items, their mean replacement values and the estimated number of damaged items, which depends on characteristics of the flood event and the affected buildings. The paper also adds insights on the model's performance and validation using two real world data sets. So, I think the paper and the model presented provide a valuable contribution to the scientific literature on flood loss modelling. Still, the paper could be further improved with regard to the following aspects.*

**Reply**: We thank the Reviewer for the positive evaluation of our work and for recognizing the contribution of the INSYDE-content model to the advancement of flood loss modelling. We appreciate the constructive comments and suggestions, which we address point by point in the following responses and will incorporate in the revised version of the manuscript.

**R2.C2**: *Introduction (line 44 - 59): While it is acknowledged that the authors present the relevant literature, not many insights about the existing approaches are provided. Please be clearer about the weaknesses and strengths of the models mentioned. And discuss later in the paper, what your model contributes in comparison to the existing approaches.*

**Reply**: In the original manuscript, we intended to provide a concise but critical overview of existing flood damage models for building contents. In particular, we noted the limited transferability of empirical models (the ones mentioned in L44-59) due to the strong regional dependence of content types and distributions (L74-76). We also discussed the only existing Italian empirical model (Carisi et al., 2018), noting that it estimates content damage indirectly via a regression on building damage. As stated in the manuscript, this approach overlooks key factors such as the spatial distribution of contents within buildings, thereby limiting its ability to capture content-specific vulnerability. Similarly, we mentioned the synthetic model by Nofal et al. (2020), emphasizing how its simplified assumptions (such as contents placement within a hypothetical single-family wooden building) hinder real-world applicability due to the absence of content and building variability representation.
Moreover, we devoted a dedicated subsection to comparing the outputs of INSYDE-content with those of Carisi et al. (2018) to empirically illustrate the differences in model behavior.
We hope that the clarifications provided above demonstrate that the current manuscript already addresses these aspects in both the Introduction and the comparative analysis. Therefore, we respectfully consider the present discussion to be adequate in its current form.

**R2.C3**: *Methodology: Since the model development is an important part of the whole paper, I think it should be presented in more detail. It doesn't become clear in my view, why these 11 items (lines 100-102) were selected.*

**Reply**: As mentioned in the original manuscript, the selection of the 11 household content items was based on a survey of real cases, conducted through the analysis of real estate listings with photographic evidence. These items represent the essential and most frequently occurring contents typically found in residential dwellings, and were selected to ensure relevance, generalizability and applicability of the model to a wide range of household types. We will clarify this point in the revised manuscript.

**R2.C4**: *Later, the sampling procedure that led to 60 buildings and the sample itself could be better described.*

**Reply**: The sample of 60 buildings was derived by applying strict selection criteria to a broader dataset of approximately 500 real estate listings examined. The selection was based on the completeness and consistency of the available information, including key geometrical attributes, architectural layouts, interior characteristics and sufficient photographic documentation to enable full model parameterization. Only listings that met all these requirements were retained, in order to ensure the reliability of the input data used in the model setup. As this point is already covered in L111-118 of the original manuscript, we consider the existing description of the sample selection to be sufficiently comprehensive and therefore believe that no further details are available to provide.

**R2.C5**: *Furthermore, Table 3 and the analyses behind it, should be better explained.*

**Reply**: To improve clarity, the revised manuscript will include a more detailed explanation of Table 3. Additionally, following the suggestion in R2.C6, we will incorporate in the main text of the paper an example illustrating how the damage function for a specific household item was derived, which we believe will further aid readers' understanding.

**R2.C6**: *A lot of material is presented in the Supplement, but I would prefer to see at least one example how the damage function was derived for one item in the main text.*

**Reply**: We appreciate the Reviewer's suggestion and, accordingly, we will include in the main text of the revised manuscript an example how the damage function for a specific household item was derived, including its fragility curve and the dependence on relevant variables. This inclusion will also support the interpretation of Table 3 (see R2.C5).

**R2.C7**: *Along the same lines, the methods in section 2.2 could benefit from some more details on the data and the methods used.*

**Reply**: The methodology and data used in Section 2.2 follow the approach adopted in the INSYDE 2.0 model for buildings (Di Bacco et al., 2024). To improve clarity and address the Reviewer's comment, we will revise the manuscript by enhancing the description of both the sensitivity analysis and validation procedures presented in this section.

**R2.C8**: *Altogether, I think the paper could benefit from a flow chart or another image showing the different stages of the model development and evaluation as well as the data sets involved.*

**Reply**: In the revised paper, we will include a flow chart in the Introduction to provide a clear visual summary of the main stages of model development and evaluation.

**R2.C9**: *Table 1: The equation for SA is given as SA = F(SA; NF). Do you mean FA as independent variable here?*

**Reply**: We thank the Reviewer for spotting this typo. The correct formulation is indeed SA = f(FA; NF), and we will correct this in the revised manuscript.

**R2.C10**: *Table 1: Why do you use FA (instead of SA) in the equation for HU?*

**Reply**: Because FA is the independent variable, while SA is derived from FA through an empirical functional relationship (i.e., HU depends on SA implicitly).

**R2.C11**: *line 176/177: rephrase ("by the same authors" is a bit confusing here)*

**Reply**: In the revised manuscript, we will rephrase the sentence to improve clarity.

**R2.C12**: *Table 3: As already mentioned above, the rationales and analyses behind the equations in Table 3 need more explanation.*

**Reply**: This will be addressed in the revised version of the manuscript (see reply to comments R2.C5 and C6).

**R2.C13**: *lines 217: "total actualized losses" - please check term*

**Reply**: In the revised manuscript we will replace "total actualized losses" with "total losses adjusted to 2023 values".

**R2.C14**: *Figure 2: This figure and the methods behind it need more explanation in my view. Also, the two data sets should be better described in the paper (briefly, but still in more detail than is currently the case).*

**Reply**: As also noted in response to comment R2.C7, the methodology underlying the results shown in Figure 2 follows the approach developed for the INSYDE 2.0 model for buildings (Di Bacco et al., 2024). In the revised manuscript, we will improve the explanation of Section 2.2 to provide additional details regarding the data and methods used in the analysis.

**R2.C15**: *Figure 4: These results should be analyzed and discussed in much more detail. How come that the estimates for (semi-)detached house do not show much variability (in contrast to the estimates for apartments)? The authors should present more in depth analysis of these results, including common metrics for errors or model performance (RMSE, MAE etc.), and they should discuss potential weaknesses of their model. How could the model be further improved to better capture the variability of the observed damage/claims?*

**Reply**: This concern was also raised by Reviewer 1 (see comment R1.C2). We report here the same response for completeness.

As also discussed in Molinari et al. (2020), claim data at the building scale are often affected by significant uncertainty and potential bias. For this reason, traditional flood damage models, as well as INSYDE-content, are typically more reliable when applied at aggregated spatial scales, rather than at the level of individual buildings.

This intrinsic variability in the observed data makes the interpretation of building-scale validation results particularly challenging. In our original manuscript (L368-374), we emphasized the importance of providing uncertainty ranges in the predicted damage values as a way to enhance the informative content of the model compared to purely deterministic approaches.

Nonetheless, we acknowledge the Reviewer's request for a clearer assessment of model performance. In the revised version of the manuscript, we will include standard error metrics (calculated based on the median of the predicted values per building, compared to the observed losses) to complement the visual inspection of the plots. However, we will also add a caveat to caution against over-interpreting these metrics, since observed values should not be considered absolute ground truth in damage modeling, due to the inherent limitations in claim data quality.

As also noted by the Reviewer, the apparent flatness of the predicted damage values for detached and semi-detached houses is partially explained by the use of log-log axes, which compress the visual perception of variability. Furthermore, from a theoretical perspective, a limited dispersion in predicted losses is to be expected for these building types, given their relatively homogeneous characteristics and the shallow inundation depths recorded during the two flood events. In contrast, greater variability is observed for multi-family residential buildings, which generally exhibit broader heterogeneity in both exposure and vulnerability, leading to capturing a damage prediction variability of the same order of that for observed losses. In the revised version of the manuscript, we will therefore include comments on these aspects regarding the interpretation of the results in Figure 4.

**R2.C16**: *In general, I think that the results could be better interpreted and discussed. In a merged section "results and discussion" there's always the risk that the discussion is too short. The authors should expand theirs.*

**Reply**: The decision to merge the results and discussion was made to ensure a more concise and coherent presentation, avoiding unnecessary repetition across sections. We believe that the key findings and their implications have already been discussed in sufficient detail throughout the combined "Results and discussion" section, with no additional elements identified that would meaningfully enrich the discussion and justify a stand-alone section.

**R2.C17**: *Supplement 1 is very helpful and detailed. It will enable others to apply the model, too, which is much appreciated.*

**Reply**: Thank you.

---

## Author Response (AR2)

**Editor**

Dear authors,

Thank you for the re-submission of your paper "INSYDE-content: a synthetic, multi-variable flood damage model for household contents" to NHESS.

As you know, two reviewers have now reviewed your revised manuscript. As you will see from the comments, in general they are pleased with the improvements made. However, two points remain (one from each reviewer) which should be addressed, namely:

**R1: Comment regarding figure 1: the reviewer and I consider this a minor comment. I would request you to try to implement the changes suggested, after which I will take a decision based on my own interpretation.**

**R2: Comment regarding the validation: the reviewer and I consider this a major comment. I request you to address this comment, which I believe is very important to this review process. After this, I will request another review of this specific section by the reviewer.**

I look forward to seeing the next version of your manuscript which I will then send out for further review as summarised above.

Please be aware, that this is most likely the last possibility for you to change and improve the manuscript. Thus, I suggest that you carefully go through the manuscript again and improve everything which you still find useful to improve (even if the referees have not pointed it out).

**Reply:** Dear Editor, we would like to thank you for your careful consideration of our revised manuscript and we are grateful for the constructive feedback provided by both reviewers and for your guidance on the two points that required further attention.

In response to Reviewer 1's comment regarding Figure 1, we have clarified the presentation of the workflow by keeping the overview figure in the main manuscript and adding a new supplementary figure (Figure S2). This figure provides a detailed schematic representation of the datasets used and the methodological steps in which they are involved. We chose to include it in the Supplement to avoid overloading the main paper, while ensuring that readers have access to a more in-depth illustration of the workflow.

Regarding Reviewer 2's major comment on the validation, we refined the discussion in this section, where we introduced a new supplementary figure (Figure S7) showing the relationship between observed content losses and inundation depth at the building level for the two case studies. This figure clearly illustrates the large, non-physical variability in the claim data, highlighting the limits of using such data as a reference for validation. We also clarified our rationale for reporting only one error metric (considering also that Figure 4 also provides information of the scatter between estimated and observed values), in order to avoid giving the misleading impression that observed claims provide an unquestionable reference. The revised text now further emphasizes that validation results must be interpreted cautiously, as they partly reflect the inherent uncertainty of the claim data rather than the model itself. Together, these revisions provide a clearer and more balanced picture of both the strengths of the model and the limitations of available observational data.

We hope that these revisions satisfactorily address the reviewers' concerns and your request.

**Reviewer #1**

**R1.C1**: Dear authors, thank you very much for the revised manuscript. The paper has improved. Although not all of my comments were entirely addressed I am satisfied with the changes made except for one thing: The new Figure 1 looks nice, but is too general and doesn't show what I was hoping for. In the method section you are describing many different steps of the data analysis and you combine and use several datasets. I was asking for a figure that depicts that workflow. Thank you for considering my comments.

Reply: We thank the reviewer for the clarification regarding the expected content of Figure 1. In the revised version of the manuscript, we decided to retain the original Figure 1 to provide a general overview of the study framework, while complementing it with a new figure (Figure S2) in the revised Supplement 1. This additional figure offers a more detailed representation of the datasets involved and the methodological steps in which they are used. We chose to include this material in the Supplement rather than the main text to avoid overloading the manuscript, and also because of its proximity to the subsequent supplementary figures that illustrate the pairplots of the two synthetic datasets (Po and extended dataset, Figures S3 and S4) and the validation datasets (Figures S5 and S6, added in this revision stage (see also our response to R2.C1)).

**Reviewer #2**

R2.C1: The manuscript has greatly improved. My main concern, the validation section, is still a problem I think. I'm particularly troubled by figure 4, as it raises more questions than it answers, it seems like there is little variation in the predictions and looking at the figure it seems that the model can really not handle the natural variation seen in the loss observations. The authors seem to agree that the log-log nature of the figure may distort it but in that case I think maybe a different figure is needed as the current figure kind of tells me the model isn't adding much value. So we would need a different figure where the log-log doesn't distort the relationship or we would need a more detailed analysis of whether the model is adding value beyond a simple constant for content loss (ie an option could also be to remove the figure and replace it by something else that answers that question). The MAE is a start but on its own doesn't provide much information on how much the complex model adds to prediction quality as we have no reference model. Other metrics like correlation or R2 might be more relevant for that question. So as a reader I would like to know whether this model is useful and currently the validation section isn't answering that for individual buildings.

That being said, it is a well known problem that flood loss models have trouble at individual building level so I'm not totally surprised by the problem and I don't think its necessarily a problem for publication if addressed properly. I just think currently the validation section raises more questions than it answers.

**Reply:** We thank the reviewer for the constructive comment, which helped us to further clarify the validation section. Indeed, validation of flood damage models at the individual-building level often raises complex issues. As also discussed in the revised manuscript, claim data should not be considered an unequivocal ground truth, as they often display inconsistencies that cannot be explained by physical processes. To explain this, we complemented our previous analysis with a new supplementary figure (Figure S7), showing the relationship between observed content losses and inundation depth at the building level for the two considered flood events. This figure clearly illustrates the large, non-physical variability in the claim data, with differences of up to an order of magnitude even for buildings of the same type and size affected by nearly identical flood conditions. Such discrepancies are likely driven by subjective household behavior or reporting practices, rather than by flood impacts

themselves, and therefore cannot be reproduced by physically informed models, such as INSYDEcontent.

For this reason, while we acknowledge the reviewer's suggestion regarding the use of additional error metrics, we deliberately decided to report only the MAE, and purely as an illustrative indicator (R² values are very low and this is clearly visible from Figure 4). Reporting a broader set of error metrics could give the misleading impression that the claim data represent an unquestionable benchmark, which is not the case, as discussed earlier. Moreover, given the substantial non-physical variability in the claim data, high values of accuracy indicators would essentially reflect how well the model reproduces the "noise" in the observation, rather than its ability to capture physically meaningful damage mechanisms.

Instead, we chose to emphasize that validation results must be interpreted with caution, since they reflect not only the model's performance but also the substantial uncertainties and inconsistencies in the reference data. We believe that the revised validation section, together with the new supplementary material, now provides a clearer picture of both the strengths of the model and the limitations of available observations, thereby addressing the reviewer's concern.

Regarding Figure 4, we believe that the log-log representation remains the most appropriate choice, especially considering the large variability in content loss estimates for apartment buildings. This visualization conveys the expected spread of losses within the plausible physical range, in line with the stage-damage functions exemplified in Figure 1, which display a certain variation in output values for similar input conditions, as in the validation cases.